# Effect of a pay-it-forward strategy on reducing HPV vaccine delay and increasing uptake among 15- to 18-year-old girls in China: A randomized controlled trial

Jing Li[1], Yifan Li[2], Chuanyu Qin[1], Yu He[3], Haidong Lu[4], Yewei Xie[5], Jason J. Ong[6], Yajiao Lu[1], Ying Yang[1], Fan Yang[7], Heng Du[8], Wenfeng Gong[8], Fei Zou[9], Heidi J. Larson[10], Mark Jit[11], Leesa Lin[10], Jennifer S. Smith[12], Elvin H. Geng[13], Dong Xu[14], Weiming Tang[15‡], Shenglan Tang[16‡], Joseph D. Tucker[6,15‡], Dan Wu[2‡*]

1 West China School of Public Health and West China Fourth Hospital, Sichuan University, Chengdu, China, 2 National Vaccine Innovation Platform & State Key Laboratory of Reproductive Medicine and Offspring Health & Department of Social Medicine and Health Education, School of Public Health, Nanjing Medical University, Nanjing, China, 3 Yulin Community Health Service Center, Chengdu, China, 4 Department of Internal Medicine, Yale School of Medicine, New Haven, Connecticut, United States of America, 5 Programme in Health Services and Systems Research, Duke-NUS Medical School, Singapore, Singapore, 6 Faculty of Infectious and Tropical Diseases, London School of Hygiene & Tropical Medicine, London, United Kingdom, 7 Institute of Population Research, Peking University, Beijing, China, 8 Bill and Melinda Gates Foundation, Beijing, China, 9 Department of Biostatistics, University of North Carolina at Chapel Hill, Chapel Hill, North Carolina, United States of America, 10 Department of Infectious Disease Epidemiology, London School of Hygiene & Tropical Medicine, London, United Kingdom, 11 Centre for Mathematical Modelling of Infectious Diseases, London School of Hygiene & Tropical Medicine, London, United Kingdom, 12 Department of Epidemiology, UNC Gillings School of Global Public Health, Chapel Hill, North Carolina, United States of America, 13 Division of Infectious Diseases, Washington University School of Medicine in St. Louis, St. Louis, Missouri, United States of America, 14 School of Health Management, Southern Medical University, Guangzhou, China, 15 Global Health Research Centre, School of Medicine, University of North Carolina at Chapel Hill, Chapel Hill, North Carolina, United States of America, 16 Duke Kunshan University, Jiangsu, China

‡ These authors are co-senior authors on this work.
* danwu@njmu.edu.cn

## Abstract

### Background

Catch-up human papillomavirus (HPV) vaccination is challenging in many low- and middle-income countries (LMICs). Pay-it-forward offers an individual a subsidized vaccine, then an opportunity to donate to help others access vaccinations. Our randomized control trial assessed the effectiveness of pay-it-forward in improving HPV vaccination among girls aged 15–18 years in China.

### Methods and findings

This study was conducted from July 6, 2022, to June 9, 2023, in four community health centers (CHCs) in Chengdu, western China. Eligible participants were unvaccinated girls living in the service areas of CHCs. Participants were initially recruited

**Data availability statement:** Anonymous data with code book and data analysis code used in the main analysis are available in the Supporting information files (Appendix C and Appendix D in S1 File).

**Funding:** The Bill & Melinda Gates Foundation (INV-034554, INV-003174 to DW, JL and ST) (https://www.gatesfoundation.org/), National Natural Science Foundation of China (82473742 to DW) (https://www.nsfc.gov.cn/english/site_1/index.html), US NIH (NIAID R01AI158826 to JDT) (https://www.nih.gov/), Nanjing Medical University Career Development Grant (NMUR20230008 to DW) (https://english.njmu.edu.cn/), Jiangsu Provincial Professorship Career Development Grant (KY103R202309 to DW) (https://www.jiangsu.gov.cn/art/2022/1/4/art_46144_10283684.html) funded this study. The study protocol underwent review by independent reviewers appointed by the funder prior to grant allocation. The funders had no role in study design, data collection and analysis, decision to publish, or preparation of the manuscript. The authors are fully responsible for the content of this manuscript, and the views and opinions described in the publication reflect solely those of the authors.

**Competing interests:** I have read the journal's policy and the authors of this manuscript have the following competing interests: EHG is an Academic Editor on PLOS Medicine's editorial board.

**Abbreviations:** CHCs, community health centers; HPV, human papillomavirus; LMICs, low- and middle-income countries; RCT, randomized controlled trial.

via telephone and, after providing verbal consent, attended in-person visit where they were randomly assigned using the sealed envelope method to either the pay-it-forward arm (received a community subsidy of 47.7 USD covering the first vaccine and an opportunity to support others) or control arm (self-paid vaccination at the market price). Participants were unblinded only after the envelope was opened, while the CHC staff coordinators, physicians prescribing the vaccine, outcome assessors, and data analysts were blinded to the intervention allocation. The primary outcome was the first-dose HPV vaccination rate, verified against clinical records. Data were analyzed using the intention-to-treat approach. We identified 662 participants per phone invitation. A total of 321 participants showed up in the health centers and randomly assigned to the pay-it-forward arm ($n=161$) or control arm ($n=160$). Most caregivers were female (80.1%, 257/321). In the pay-it-forward arm, 55 of 161 (34.2%) girls received the HPV vaccine, compared with 28 of 160 (17.5%) girls in the control arm (adjusted proportion difference = 17.9%, (95% CI [8.7%, 27.0%]; $P<0.001$). Among 55 girls in the pay-it-forward arm who received the vaccination, 37 (67.3%) wrote a postcard message, and 39 (70.9%) of their caregivers donated to support future girls. The financial cost per person vaccinated was $294 in the control arm and $230 in the pay-it-forward arm. The trial had several limitations, including a 54% clinic attendance rate (360 of 662 consented participants attended) and its conduct in a single western province of China.

## Conclusions

The pro-social pay-it-forward strategy was effective to increase catch-up HPV vaccination among teenage girls. This approach also enhanced vaccine confidence among participants. Pay-it-forward demonstrates promise as an effective intervention to improve vaccine uptake through community engagement.

## Trial registration

Chinese clinical trial registry ChiCTR2200055542 (https://www.chictr.org.cn/hvshow-project.html?id=183292&v=1.3).

---

### Author summary

### Why was this study done?

- In China, subsidized HPV vaccination programs mainly target girls aged 13–14, leaving catch up age girls (15–18 years) with low vaccination rates.

- Many girls face barriers such as low awareness, misinformation, and vaccine hesitancy, and community support is essential to address these issues.

- This study tested a community-engaged pay-it-forward model that offers subsidized vaccines and encourages recipients to support others through donations and messages.

## What did the researchers do and find?

- We conducted a randomized trial involving 321 catch-up age girls and their caregivers in community health centers in Chengdu, western China.
- The pay-it-forward strategy significantly increased HPV vaccine uptake, improved vaccine confidence, and reduced vaccination delay compared to control arm.
- Most vaccinated participants in the pay-it-forward arm wrote encouraging messages and many caregivers donated to help future recipients. The cost per person vaccinated was lower than in the control group.

## What do these findings mean?

- The pay-it-forward model is a promising and cost-effective way to boost HPV vaccination among older adolescent girls through community engagement.
- However, this study was conducted in one province in western China.
- More research is needed to explore how this prosocial approach can support other health programs in different settings.

## Introduction

Catch-up human papillomavirus (HPV) vaccination is challenging, and barriers exist in many low and middle-income countries (LMICs). Despite the clear benefits of free HPV vaccines, many health systems still impose fees. The cost of a single dose of HPV vaccine is comparable to a young woman's earnings over several months in LMICs. Such costs hinder vaccine uptake, especially among the most vulnerable [1], erode trust in healthcare [2], and impede universal health coverage goals. Providing free HPV vaccines could accelerate the achievement of HPV vaccine uptake goals. However, countries where free HPV vaccination is available have suffered from low uptake [3], indicating other barriers in addition to fees. Lack of awareness and vaccine hesitancy may be important drivers in these settings [4]. Misinformation can also undermine confidence in vaccines. Public engagement is essential for tackling such barriers and gaining community acceptance of HPV vaccination [5,6], but effective community-engaged programs are limited [7]. Most evaluations of HPV interventions have been conducted in high-income countries, with relatively few studies carried out in middle-income countries [8].

Some local governments in China have introduced subsidized HPV vaccination programs for girls aged 13–14, but these exclude catch-up vaccinations for older girls (15–18), leaving low coverage rates (<15%) among this age group [9,10]. To achieve the cervical cancer elimination goal, catch-up age groups should also be included [11]. Budget constraints prohibit government's extension to older age groups, and women are unable or reluctant to self-pay for the widely available vaccine. In China, two doses of the HPV vaccine are recommended for girls aged 9–14, while three doses are required for girls in the catch-up age group.

Earlier HPV vaccination is recommended for several benefits, i.e., better protection effects, fewer doses, and lower costs [12]. However, vaccine delay—defined as receiving the vaccination after the recommended age—increases the risk of HPV infection [13] and remains a significant barrier, even among many financially capable individuals [14]. Evidence shows that over 61% of Chinese caregivers had a high level of willingness to vaccinate their girls [15], but 80% [16] choose to delay vaccination at a later age due to poor awareness, assuming that girls under 18 face minimal risk

since they are mostly at home or in school [14]. A novel community-engaged solution is urgently needed to address the above financial and perception-related challenges. Community engagement involves building partnerships that empower stakeholders to work together in addressing health challenges and promoting overall well-being [17]. Community-engaged interventions have been shown to be effective in increasing the vaccine uptake [18].

Pay-it-forward, a community-supported subsidy approach that combines incentive and priming nudges [19], may provide such an opportunity. It gives a person an incentive nudge (e.g., subsidized vaccine) and a kindness-based priming nudge (e.g., community handwritten postcard), and then offers the person an opportunity to give back to the community by donating money and/or creating HPV vaccine promotion messages (interpersonal affecting nudging) (Fig A in S1 File) [20]. Pay-it-forward builds on the theory of upstream reciprocity (i.e., individuals helped by someone are more likely to help others) [21]. Our earlier research demonstrated that pay-it-forward increased health services uptake among priority populations by generating community engagement and enhancing participant confidence in vaccine importance, safety, and effectiveness [22–24]. In this two-arm randomized controlled trial (RCT), we aimed to evaluate whether the community-engaged pay-it-forward strategy would reduce vaccine delay behavior and increase earlier HPV vaccine uptake against the standard self-paid vaccination among girls aged 15–18 years in Chengdu, western China where a lower HPV vaccine uptake was observed (8.6%) as compared to other regions [9,10].

## Methods

### Study design and participants

This was a two-arm RCT conducted in Chengdu, Sichuan Province, China. Chengdu had a population of 21 million people in 2022. Chengdu is located in western China, a less developed region with significant economic disparities, and itself exhibits uneven development, containing both affluent and underdeveloped areas. The trial was organized at four community health centers that provide routine vaccination services. The protocol has been published (Protocol in S2 File) [25]. Four community health centers were selected because they had the following: essential infrastructure for providing HPV vaccination services, including personnel, and stable HPV vaccine supply. The four sites by relative level of individual disposable income in 2021 were Site A (most developed urban area), Site B (higher middle-income suburban area), Site C (lower middle-income suburban area), and Site D (least developed area) [26]. The Ethics Committee of West China Fourth Hospital and West China School of Public Health approved the study (Gwll2021057/ Gwll2023125). This trial had been registered on Chinese Clinical Trial Registry and WHO International Clinical Trials Registry Platform (ChiCTR2200055542). The trial URL is https://trialsearch.who.int/Trial2.aspx?TrialID=ChiCTR2200055542. We used CONSORT guidelines [27] and CONSERVE (for this trial was affected by the COVID-19 pandemic) [28] for reporting this RCT trial (S1 and S2 Checklist files).

### Participant recruitment and screening

We used the simple random sampling method to select girls. The complete name list of adolescent girls aged 15–18 years in the study area was obtained from the neighborhood committee via community health center staff. This list included all girls of this age range living in the neighborhood who registered their names and contacts of caregivers at the community health center to receive relevant health services, including vaccines. We assigned a random number to each individual using Microsoft Excel. Individuals were sorted by random numbers from the smallest to the largest, and we selected potential participants from the top of the list for telephone recruitment until we reached the sample size of 80 eligible girls at each study site.

Telephone recruitment included a brief introduction to the overall study purpose, procedures, and associated benefits and risks (Appendix A in S1 File). Inclusion criteria for participants were (1) adolescent girls aged between 15 and 18 years with a caregiver, (2) living in the areas that the chosen community health centers serve, (3) no history of HPV vaccination, and (4) no known vaccine allergy. Caregivers of girls confirmed eligibility and were then invited with their girls to the community health center. Participating caregivers provided verbal consent during telephone recruitment and

signed a written informed consent form, and girls provided assent during the on-site visit. Recruitment occurred over a prolonged eight-month period because of COVID-19 lockdowns and community re-direction of vaccine services to prioritize COVID-19 vaccination. Site D recruitment coincided with the relaxation of COVID-19 policies when many clinics were overwhelmed with COVID-19 and vaccine sites were short-staffed (Fig B in S1 File).

### Randomization and masking

Simple randomization of participants was used. An independent statistician prepared the computer-generated randomization list, and we used the sealed envelope method to randomize eligible participants (caregiver-girl pairs) into control and pay-it-forward arms at a 1:1 ratio within each center, based on the order in which they visited the sites. An individual-based randomization approach was appropriate for efficacy evaluation of a newly developed strategy and is more cost-effective [29]. The community health staff coordinator, physicians who prescribed HPV vaccines, the outcome assessors, and data analysts were blinded to intervention allocation. Additionally, the following techniques were adopted to ensure the concealment of allocation in this trial: the process of generating the sequence of random digits and assembling envelopes was confidential to all on-site research staff who recruited participants; the allocated intervention was concealed from both participants and research staff onsite prior to assignment until they opened the envelope.

### Procedures

Caregiver-girl pairs in the control arm received a routine educational pamphlet used in clinics which included the prices of available HPV vaccines. They had to pay out of pocket at the standard price if they wanted to receive HPV vaccination.

In addition to the routine educational pamphlet, caregiver-girl pairs in the pay-it-forward arm received the intervention that included community co-created postcard messages, featuring messages from previous participants on the back (Fig C in S1 File), subsidized HPV vaccination (covering RMB 329, equivalent to 47.70 USD), and the opportunity to voluntarily donate towards someone else's vaccine dose and/or write postcard messages. Donations were integrated into the program's operational budget and used to support future HPV vaccinations for eligible girls in the same community. Participants in the pay-it-forward arm were told the market prices to receive available HPV vaccines, and that previous participants had donated RMB 329 (47.7 USD, equal the price of the first dose of domestic HPV vaccination) towards the price, along with postcard messages for them. They were also told that other adolescent girls from their neighborhood cared about them. We used a series of community engagement methods, including stakeholder meetings ($n = 18$), participant interviews and focus group discussions ($n = 21$), postcard messages co-designed with college students ($n = 4$), to adapt and refine the intervention components taking population characteristics and contextual factors into consideration (Appendix B in S1 File). Participants in both arms were given a brief introduction to the project, pamphlet information about cervical cancer, HPV vaccination details, and the price of HPV vaccine (Fig D in S1 File).

Caregivers who agreed to have their daughter receive a vaccine made an appointment for their daughter to receive vaccination within three months. In the pay-it-forward arm, during the half-hour post-vaccine monitoring period, girls received subsidized vaccination and handwritten postcards, and they had an opportunity to create their own postcard messages for others, while caregivers were asked about donation. A donation/postcard collection box was provided on-site, and QR code was provided to those who preferred online donation. The donation was voluntary and anonymous. Project staff left participants after bringing them the postcards, and they were unaware of participants' donation behavior and donation amount. Our earlier pilot trial demonstrated the intervention process was acceptable and well-received by participant [30].

We used information tracking sheet to collect data about vaccine uptake and adverse events within 24 h, which were verified against clinical records. All participating caregivers completed an online questionnaire (Appendix C in S1 File) on demographic information and vaccine attitudes after they received the project information and intervention materials. The

number of participants who donated, and the donation amount in the pay-it-forward arm were verified against the official project account hosted on WeChat (a mobile app with monetary transaction function). The cost data were obtained from project expense records and by asking local health staff.

## Outcomes

The primary outcome of the study was first-dose HPV vaccine uptake within three months after randomization and ascertained by clinical records. The four community health centers had a stable supply of bivalent HPV vaccine (Cecolin) during the study period.

Secondary outcomes included second-dose HPV vaccine uptake, number of pay-it-forward participants who donated, donation amounts received, the costs for implementing the intervention and cost per person vaccinated from the healthcare provider perspective, vaccine confidence, and vaccine delay [13]. Information on second-dose HPV vaccine uptake was collected from clinical records, with no additional intervention administered after the first dose. Vaccine confidence was measured using standard survey items adapted to assess HPV vaccine confidence in China [31]. Vaccine delay refers to individuals have positive intentions to vaccinate but fail to act [32]. The measurement of vaccine delay was adapted from a study evaluating COVID-19 vaccine delay [33]. We asked participants' willingness to receive the vaccine, and the actual vaccination was verified using clinical records after the 3-month follow-up period. If participants expressed willingness to receive the vaccine but did not obtain it by the end of the follow-up period, this was operationally defined as vaccine delay.

## Statistical analysis

Details about the sample size calculation are available elsewhere [25]. In short, based on current vaccination rates of <15% [9,10] and the proportion difference of 16% between the two arms in our pilot study [25], we estimated that the outcome vaccination rate of the control arm was 20% due to the provision of routine educational pamphlet, and 36% in the pay-it-forward arm. Assuming a two-sided Z-Test with unpooled variance, a sample size of 240 (or 120 in each arm) can achieve 80% power in detecting differences between the two arms with a significance level of 0.05. Given a non-response and/or dropout rate of 10%, 133 participants for each arm were needed. To allow sub-analyses on secondary outcomes and sub-group analyses, we increased the sample size by 20% to 160 in each arm. Based on the structure of our study, with four centers participating, the sample size of 160 per arm corresponds to approximately 80 participants per center (40 in the control arm and 40 in the intervention arm).

We followed a preregistered analysis plan and Protocol in S2 File clarified amendments. Descriptive analyses of socio-demographic characteristics and vaccination rates by study arms were conducted. The pay-it-forward and control arms' uptake rates were compared using chi-squared tests. Crude and adjusted proportional differences in HPV vaccine uptake comparing the pay-it-forward arm with control arm were estimated using generalized linear models with Gaussian distribution and identity link function, and the corresponding 95% confidence intervals were calculated using sandwich variance estimator [34]. The adjusted proportional differences accounted for study sites, household income level, education level, marital status, and sex of the caregiver. The secondary outcomes—vaccine confidence and vaccine delay—were summarized using descriptive statistics. Similarly, comparisons of these secondary outcomes between the two arms were also assessed using generalized linear models with Gaussian distribution and identity link function. Difference in donation amount between different sites in the pay-it-forward arm was assessed using independent-Samples Kruskal–Wallis Test.

We employed a micro-costing method to evaluate the financial cost of implementing HPV vaccination under the two intervention strategies from the health provider's perspective (Sichuan Department of Health). We tracked all resources used during the experiment and classified cost items. The collected cost data and costs-related results are in the Table A in S1 File. We presented the total financial cost for each group, the cost per person vaccinated, and the donation in the

pay-it-forward group. All expenses are expressed in 2022 USD using OANDA currency conversions (1 USD = 6.89 RMB, year of 2022).

Subgroup analyses were pre-specified to assess whether the intervention effects on vaccine uptake varied by (1) household income levels (≤11,611 USD/year or > 11,611 USD/year). This threshold was determined based on the mean household income in rural Sichuan Province in 2021; [35] (2) study sites A, B, C, D; (3) sex of the participating caregiver (female or male); (4) highest educational attainment of the participating caregiver (college and above, or high school and below); (5) age of caregivers (<40 or ≥40 years old); (6) awareness of HPV vaccine (yes or no). Both crude and adjusted proportional differences in HPV vaccine uptake comparing the pay-it-forward arm with control arm for each subgroup were reported. To test for heterogeneity of the intervention effect across subgroups, we included interaction terms between the intervention group and each subgroup variable in the regression models. Data were exported to R Statistical Software (version 4.2.1; R Foundation for Statistical Computing, Vienna, Austria; Appendix D in S1 File) for analysis. We conducted the cost analysis in Excel 2019 (Microsoft, USA).

The leading academic research organization (Nanjing Medical University, NMU) monitored the study. No data safety monitoring committee was appointed. This study was registered with ChiCTR (ChiCTR2200055542), and is now closed for recruitment.

## Results

From July 6, 2022, through June 9, 2023, a total of 662 eligible caregiver/daughter pairs verbally agreed to attend clinics via telephone recruitment, and 360 were present for eligibility screening at clinics (clinic response rate = 54%) (Fig 1). Overall, 39 were further excluded due to age ineligibility (n = 1), decision to receive HPV vaccination via other channels (n = 21) or refuse to consent to participate (n = 17), and 321 were included in the final data analysis: 160 were assigned to the control arm, and 161 were assigned to the pay-it-forward arm.

The baseline characteristics were similar with no statistical difference between the two arms (Table 1). Most caregivers were female (257/321, 80.1%), married (258/321, 80.4%), reported a household income level of ≤11,611 USD/year (233/321, 72.6%), and had an educational background of high school or less (231/321, 72.0%). Most participants (291/321, 90.6%) reported no family history of HPV or cervical cancer diagnosis.

A total of 32.1% (103/321) of participants scheduled appointments, with 42.2% (68/161) in the pay-it-forward arm, and 21.9% (35/160) in the control arm (Fig 1; Table B in S1 File). During the follow-up period, a total of 25.9% (83/321) received the vaccination, with 34.2% (55/161) in the pay-it-forward arm, and 17.5% (28/160) in the control arm (Appendix E in S1 File). Pay-it-forward participants were significantly more likely to receive a vaccination than those in the control arm, with an adjusted proportional difference of 17.9% (95% CI [8.7%, 27.0%]) (Fig 2). No severe adverse events within 24 hours following the vaccination were reported.

Pre-specified subgroup analyses (Fig 2) showed that pay-it-forward was associated with a significant increase in vaccine uptake among girls (1) when female caregivers were present for the intervention (adjusted proportion difference = 20.2% (95% CI [9.7%, 30.7%]), (2) among caregivers aged 40 or above (adjusted proportional difference = 18.4% (95% CI [7.6%, 29.2%]), (3) across education levels (high school or below: adjusted proportion difference = 16.3% (95% CI [4.9%, 27.7%]); college or above: adjusted proportion difference = 28.4% (95% CI [10.7%, 46.2%]), (4) among caregivers with lower household incomes (adjusted proportion difference = 19.8% (95% CI [8.9%, 30.1%]), (5) among caregivers who were aware of HPV vaccines before participating this study(adjusted proportion difference = 20.2% (95% CI [10.9%, 29.5%]), and (6) across two study sites (Site A: adjusted proportion difference = adjusted proportion difference = 22.8% (95% CI [6.7%, 38.9%]); Site C: adjusted proportion difference = 25.4% (95% CI [4.3%, 46.6%]). Interaction p-values are reported in Table 2 to assess heterogeneity of intervention effects across subgroups.

Among 83 girls who received the first dose vaccine, 69 of them received the second dose, with the second-dose vaccine uptake rates being 89.1% (49/55) and 71.4% (20/28), respectively, in the pay-it-forward and control arms (Fig 3).

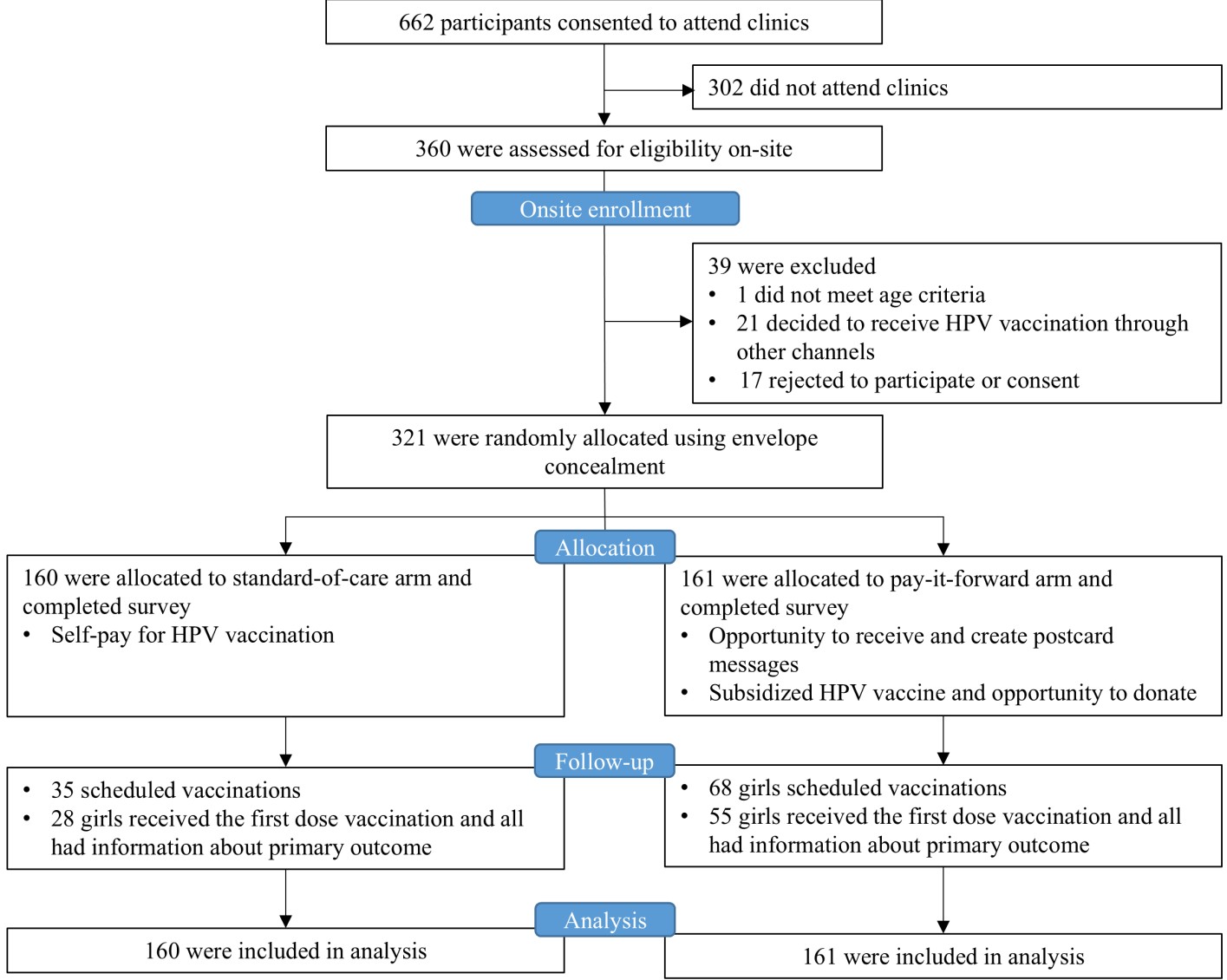

**Fig 1. Trial profile and CONSORT diagram.**

Our analysis of vaccine confidence (Fig 3) showed that pay-it-forward participants reported significantly higher levels of confidence in vaccine importance, safety, and effectiveness, for both nonavalent and bivalent HPV vaccines than those in the control arm. A lower proportion of participants in the pay-it-forward arm showed vaccine delay compared to that in the control arm (adjusted proportion difference = −18.4% (95% CI [−27.6%, −9.2%]; Fig 3).

The total financial cost for the healthcare provider of implementing an HPV vaccination intervention for participants was $8,240 for the control arm, and $12,676 for the pay-it-forward arm. Among participants in the pay-it-forward arm who received vaccination, 70.9% of participants (39/55) donated money, and the total donation amount was $332.2, with a median of $2.9 (IQR 1.6–9.6). The donation distribution by study site is shown in Table 2 and Fig E in S1 File. The financial cost per person vaccinated was $294 in the control arm and $230 in the pay-it-forward arm. The proportion of financial cost related to variable cost was 36.7% and 42.7% in the control and pay-it-forward arm, respectively (Fig F in S1 File).

**Table 1. Characteristics of the girls 15–18 years old and their caregivers in Chengdu, China, 2022-2023, n=321.**

| | Control (n=160) | Pay-it-forward (n=161) | Total (n=321) |
|---|---|---|---|
| Age (girl)—years | 16.6(0.9) | 16.6(1.0) | 16.6(1.0) |
| Age (caregiver)—years | 44.4(8.6) | 43.0(8.5) | 43.7(8.6) |
| Sex of caregiver | | | |
| Male | 26(16.3) | 38(23.6) | 64(19.9) |
| Female | 134(83.7) | 123(76.4) | 257(80.1) |
| Education level of caregiver | | | |
| High school or less | 114(71.3) | 117(72.7) | 231(72.0) |
| College or more | 46(28.7) | 44(27.3) | 90(28.0) |
| Marital status of caregiver | | | |
| Unmarried | 14(8.8) | 23(14.3) | 37(11.5) |
| Married | 134(83.7) | 124(77.0) | 258(80.4) |
| Divorced or widowed | 12(7.5) | 14(8.7) | 26(8.1) |
| Household income levels[a] | | | |
| ≤11,611 USD/year | 110(68.8) | 123(76.4) | 233(72.6) |
| >11,611 USD/year | 50(31.2) | 38(23.6) | 88(27.4) |
| Awareness of HPV vaccines | | | |
| Yes | 146(91.3) | 140(87.0) | 286(89.1) |
| No | 14(8.7) | 21(13.0) | 35(10.9) |
| Whether anyone in the same household has ever been diagnosed with HPV or cervical cancer | | | |
| Have been diagnosed | 11(6.9) | 4(2.5) | 15(4.7) |
| Have not been diagnosed | 141(88.1) | 150(93.2) | 291(90.6) |
| Not clear | 8(5.0) | 7(4.3) | 15(4.7) |
| Study sites | | | |
| Site A | 40(25.0) | 43(26.7) | 83(25.9) |
| Site B | 44(27.5) | 39(24.2) | 83(25.9) |
| Site C | 38(23.8) | 39(24.2) | 77(24.0) |
| Site D | 38(23.8) | 40(24.9) | 78(24.4) |

1USD=6.89RMB. HPV, human papilloma virus. Data are mean (SD) or n (%).

[a]The average household income in rural Sichuan Province was around RMB 80,000 in 2021 according to Sichuan Statistical Yearbook 2022.

Out of 55 girls vaccinated in the pay-it-forward arm, 37 (67.3%) girls wrote a postcard message to encourage to get vaccinated (Fig G in S1 File). Out of the total of 37 messages, 18.9% (7/37) encouraged other girls to vaccinate, and 83.8% (31/37) expressed their wish for a world free of diseases and conveyed their blessings to other girls in the format of a text message or a poem.

## Discussion

HPV vaccination is the most effective way to prevent cervical cancer, but community-engaged strategies to promote uptake are limited. Our data demonstrated that the pay-it-forward strategy can increase HPV vaccine uptake compared to the control. Participants in the intervention arm showed greater vaccine confidence, less vaccine delay, and generated community messages and engagement compared to those in the control arm. Our study extends the literature by focusing on an innovative incentive strategy for HPV vaccination, and leveraging prosocial tendencies to enhance public health. This is a rare example of a community-engaged intervention to reduce HPV vaccine delay and enhance vaccine uptake in a middle-income country context.

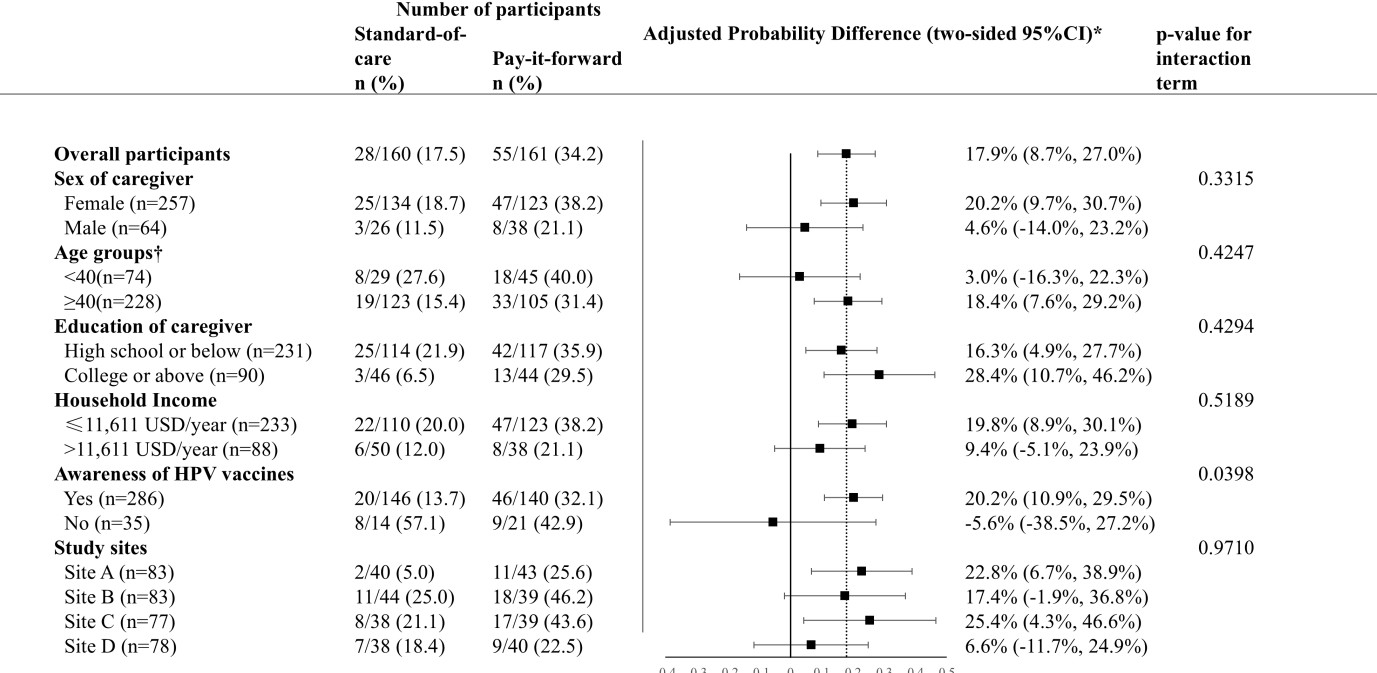

**Fig 2. Generalized linear models to compare HPV vaccine uptake rates of two arms.** 1USD = 6.89RMB. HPV, human papilloma virus. *The proportional differences were adjusted for study sites, household income level, education level and marital status and sex of the caregiver. †There are missing data of the age of caregiver.

**Table 2. Donations by study sites in the pay-it-forward arm (USD).**

|  | Vaccinated | Donated | Total donation | Median donation | Minimum donation | Maximum donation | *P*-value[a] |
|---|---|---|---|---|---|---|---|
| Site A (*n* = 83) | 11 | 11(100.0) | 134.6 | 8.7 (5.1, 19.2) | 2.4 | 29.0 | 0.082 |
| Site B (*n* = 83) | 18 | 16(88.9) | 160.6 | 5.1 (1.5, 14.9) | 0.1 | 47.9 | .. |
| Site C (*n* = 77) | 17 | 8(47.1) | 13.5 | 2.3 (1.5, 3.3) | 1.5 | 7.3 | .. |
| Site D (*n* = 78) | 9 | 4(44.4) | 23.5 | 2.9 (2.3, 4.0) | 0.4 | 7.3 | .. |
| Total | 55 | 39(70.9) | 332.2 | 2.9 (1.6, 9.6) | 0.1 | 47.9 | .. |

Data are *n*, *n*(%) or median (IQR). 1USD = 6.89RMB.

[a]Independent-Samples Kruskal–Wallis Test was used to compare differences of median donation amount between different sites.

The pay-it-forward arm had a greater vaccine uptake rate than the control. This is consistent with previous studies using pay-it-forward to improve influenza vaccination and other health services uptake [22–24]. The HPV vaccine uptake in the pay-it-forward arm was significantly higher than that among girls aged ≤18 years old in a 2021 national survey (12.4%) [9]. Pay-it-forward also had a higher uptake compared to the impact of a 2015 UK-based financial incentive intervention study on HPV vaccine uptake among girls aged 16–18 years (28.4%) [36]. Although the UK-based study was somewhat dated, China's HPV vaccine services are about 10 years behind the UK, therefore, these figures were presumably comparable. Pay-it-forward was also twice as effective among caregivers with lower household incomes compared to those with higher incomes, which might have implications for reducing HPV vaccination inequalities. The effect may be because of community-contributed financial support, locally relevant postcard messages, community engagement, or some combination [37]. However, Center D—located in the least developed area—showed no improvement. This may be

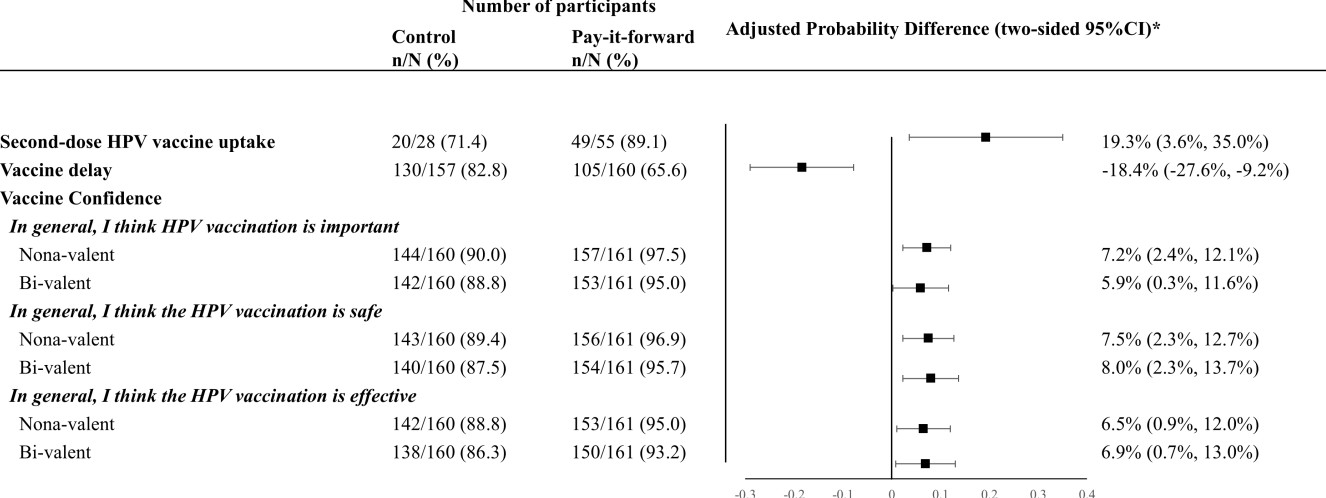

| | Number of participants | | Adjusted Probability Difference (two-sided 95%CI)* | |
| --- | --- | --- | --- | --- |
| | Control n/N (%) | Pay-it-forward n/N (%) | | |
| Second-dose HPV vaccine uptake | 20/28 (71.4) | 49/55 (89.1) | | 19.3% (3.6%, 35.0%) |
| Vaccine delay | 130/157 (82.8) | 105/160 (65.6) | | -18.4% (-27.6%, -9.2%) |
| **Vaccine Confidence** | | | | |
| *In general, I think HPV vaccination is important* | | | | |
| Nona-valent | 144/160 (90.0) | 157/161 (97.5) | | 7.2% (2.4%, 12.1%) |
| Bi-valent | 142/160 (88.8) | 153/161 (95.0) | | 5.9% (0.3%, 11.6%) |
| *In general, I think the HPV vaccination is safe* | | | | |
| Nona-valent | 143/160 (89.4) | 156/161 (96.9) | | 7.5% (2.3%, 12.7%) |
| Bi-valent | 140/160 (87.5) | 154/161 (95.7) | | 8.0% (2.3%, 13.7%) |
| *In general, I think the HPV vaccination is effective* | | | | |
| Nona-valent | 142/160 (88.8) | 153/161 (95.0) | | 6.5% (0.9%, 12.0%) |
| Bi-valent | 138/160 (86.3) | 150/161 (93.2) | | 6.9% (0.7%, 13.0%) |

**Fig 3. Generalized linear models to compare second-dose HPV vaccine uptake, vaccine delay and vaccine confidence rates of two arms.** *The proportional differences were adjusted for study sites, household income level, education level and marital status and sex of the caregiver. All secondary outcome data were collected during follow-up. HPV, human papilloma virus.

because limited awareness of the HPV vaccine, rather than financial constraints, was the primary barrier in this region, though further research is needed to confirm this. Additionally, our control arm demonstrated higher uptake (17.5%) compared to the previously reported rate of 8.6% in western China. This difference may be attributable to implementation components such as telephone recruitment reminders, educational pamphlets addressing knowledge gaps, and established trust in community health centers.

Pay-it-forward may be a novel incentive strategy and had lower costs per person vaccinated compared to the control approach. Among those recruited in the pay-it-forward arm, most participants who received a vaccine voluntarily donated to support another girl. A systematic review suggested that conventional financial incentives (e.g., cash payments, vouchers) are largely effective in increasing vaccine uptake [38], and our research adds to the literature by demonstrating the effectiveness of an innovative community-engaged incentive design. Despite small mean donations collected through the pay-it-forward strategy, we observed higher average donations in the two better-off regions than in the lower-income regions, suggesting the potential of creating a subsidization mechanism to support HPV vaccination in poorer regions. Of note, pay-it-forward is not intended to replace the government role, but rather to serve as an incentive structure or complementary model to support future catch-up vaccination programs.

The pay-it-forward intervention generated additional community-created messages, mostly written by adolescent girls, to encourage future's vaccination. Community engagement is essential to the success of health service programs. Community engagement focused on cultivating kindness may improve confidence, better acceptance of vaccine services, and our pay-it-forward participants showed significantly better vaccine confidence and less vaccine delay. Previous data showed that pay-it-forward was associated with increased community solidarity [39], and vaccine confidence [22]. A systematic review demonstrated that prosocial interventions are associated with improved health outcomes for both givers and recipients and have been implemented across diverse settings, highlighting their adaptability to various cultural and healthcare contexts [37]. Pay-it-forward may also provide a novel nudging approach for effective messaging and vaccine communication by fostering a culture of mutual support and trust [37]. This may be particularly relevant to regions where free or subsidized vaccine services are available, but have sub-optimal uptake rates due to vaccine hesitancy.

The study has some limitations. First, our study was implemented after the Chengdu municipal government rolled out the subsidized HPV vaccination program for 13–14-year-old girls, which might have improved public awareness and created additional peripheral demand among other age groups for HPV vaccines. However, all participants were randomized, and we expect this change had a similar impact on our study outcomes between the two arms. Second, our study had a clinic attendance rate of 54% and a long recruitment period. Our study was conducted during COVID-19 control measures, and there were COVID-19 cases and short-term (3–7 days) community-scale lockdowns to contain viral spread. This might have reduced participants' likelihood to visit clinics for recruitment or not showing up for appointments, resulting in underestimated response and vaccination rates in the study. Third, the pay-it-forward strategy is a multi-component intervention, and its individual effects cannot be isolated. Future research could consider designs that allow disentanglement of individual components. Finally, the study is conducted in a western province in China, and the study sites were selected based on the availability of vaccines, organizational willingness, and capacity to collaborate. Selection bias caused by convenience sampling of study sites is possible, and generalizability to the general population should be made cautiously [40]. But our study sites from different economic settings reflected common pathways for HPV vaccination in China.

Our study has important implications for practice, research, and policy in China and other LMICs. First, this is a rare community-engaged intervention to increase HPV vaccine uptake among priority populations in a middle-income country. The research findings may have important implications for pro-social interventions to improve public trust and address vaccine delay. Second, the community engagement methods to generate contextually appropriate intervention materials and successful implementation of a novel multi-component strategy in clinical settings may support evidence-based practices and effective public messaging. Third, pay-it-forward is an innovative incentive design that leverages human kindness to expand public health services and address intention-to-behavior gap. The model is highly adaptable to local contexts where giving and reciprocity culture exist, and gift components can be contextualized [37]. Fourth, it can complement government-led HPV vaccination initiatives to broaden access for more individuals and might be a transition model to universal health coverage for catch-up HPV vaccination programs. Future research can explore potential mechanisms of how pay-it-forward works, the sustainability and scalability in similar contexts. Additional studies are also needed to assess the feasibility of integrating these solutions into routine services.

## Supporting information

**S1 Checklist. CONSORT 2025 checklist of information to include when reporting a randomized trial.**
(DOCX)

**S2 Checklist. CONSERVE-CONSORT checklist.**
(DOCX)

**S1 Data. Database for pay-it-forward to reduce HPV vaccine delay and increase uptake.**
(XLSX)

**S1 File. Supporting information. Fig A**. Overview of pay-it-forward. **Fig B**. Study site recruitment over time. **Fig C**. Community engaged postcards. **Fig D**. Educational pamphlet. **Fig E**. Percentages of donations of vaccine costs by study sites. **Fig F**. Breakdown of financial costs by category in proportions in the two arms. **Fig G**: Translated example hand-written postcard messages from the participants. **Appendix A**. Telephone recruitment script. **Appendix B**. Community engagement activities. **Appendix C**. Questionnaire and codebook. **Appendix D**. Data analysis code. **Appendix E**. Video abstract. **Table A**. Cost calculations (in 2022 USD) for the study. **Table B**. Recruitment, appointment, and reasons for non-presence for vaccination.
(DOCX)

**S2 File. Protocol.** Study protocol and amendments to the protocol.
(DOCX)

## Acknowledgments

The authors are grateful to all participants. We would like to thank all collaborative health staff in Yulin Community Health Service Center, Longtan Community Health Service Center, Xinjin District Maternal and Child Healthcare Hospital, and Third People's Hospital of Chengdu Eastern New Area for their contribution to the execution of this study. We thank all advisory group members who provided advice on contextual background.

## Author contributions

**Conceptualization:** Jing Li, Fei Zou, Jennifer S. Smith, Dong Xu, Dan Wu.

**Data curation:** Yifan Li, Chuanyu Qin, Haidong Lu, Yewei Xie, Yajiao Lu, Dan Wu.

**Funding acquisition:** Jing Li.

**Investigation:** Jing Li, Yifan Li, Chuanyu Qin, Yu He, Yajiao Lu, Ying Yang, Dan Wu.

**Methodology:** Yifan Li, Chuanyu Qin, Haidong Lu, Yewei Xie, Jason J. Ong, Ying Yang, Heng Du, Wenfeng Gong, Fei Zou, Heidi J. Larson, Mark Jit, Leesa Lin, Jennifer S. Smith, Elvin H. Geng, Dong Xu, Weiming Tang, Shenglan Tang, Joseph D. Tucker, Dan Wu.

**Project administration:** Jing Li, Yu He, Dan Wu.

**Resources:** Yu He, Jason J. Ong, Yajiao Lu.

**Software:** Haidong Lu, Yewei Xie, Jason J. Ong.

**Supervision:** Fan Yang, Weiming Tang, Shenglan Tang, Joseph D. Tucker.

**Writing – original draft:** Jing Li, Yifan Li, Chuanyu Qin, Haidong Lu, Jason J. Ong, Dan Wu.

**Writing – review & editing:** Jing Li, Yifan Li, Fan Yang, Fei Zou, Heidi J. Larson, Mark Jit, Leesa Lin, Jennifer S. Smith, Elvin H. Geng, Dong Xu, Weiming Tang, Shenglan Tang, Joseph D. Tucker, Dan Wu.

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
