## [Editor Report · Decision Letter 0]

8 Jan 2025

Dear Dr Wu, 

Thank you for submitting your manuscript entitled "Pay-it-forward strategy reduced HPV vaccine delay and increased uptake among catch-up age girls: A randomized clinical trial" for consideration by PLOS Medicine.

Your manuscript has now been evaluated by the PLOS Medicine editorial staff as well as by an academic editor with relevant expertise and I am writing to let you know that we would like to send your submission out for external peer review.

We also ask that you provide a copy of the original trial protocol as supporting information. By protocol, we mean the complete and detailed plan for the conduct and analysis of the trial that the ethics committee approved before the trial began (i.e. Version 1.0). Please send this in the original language. If this is in a language other than English, please also provide a translation. Please detail any deviations from this study protocol in the Methods section of your manuscript. The documents will be made available to the editors and reviewers.

Please re-submit your manuscript within two working days, i.e. by Jan 10 2025 11:59PM.

Kind regards,

Suzanne

Suzanne De Bruijn, PhD

Senior Editor

PLOS Medicine

---

## [Decision Letter · Decision Letter 1]

22 Apr 2025

Dear Dr Wu,

Many thanks for submitting your manuscript "Pay-it-forward strategy reduced HPV vaccine delay and increased uptake among catch-up age girls: A randomized clinical trial" (PMEDICINE-D-25-00007R1) to PLOS Medicine. The paper has been reviewed by subject experts and a statistician; their comments are included below and can also be accessed here: [LINK]

As you will see, the reviewers thought this was a well-conducted trial, but also had several concerns. After discussing the paper with the editorial team and an academic editor with relevant expertise, I'm pleased to invite you to revise the paper in response to the reviewers' comments. We want you to address all reviewers concerns, in particularly the comment from Reviewer 2 concerning the fact that the monetary amounts donated are too small for this to be a 'pay-it-forward' study, and the comment from Reviewer 3 regarding the control group. Please also address the comment from the Academic Editor. We plan to send the revised paper to some or all of the original reviewers, and we cannot provide any guarantees at this stage regarding publication.

We ask that you submit your revision by May 13 2025 11:59PM. However, if this deadline is not feasible, please contact me by email, and we can discuss a suitable alternative.

Don't hesitate to contact me directly with any questions (sbruijn@plos.org). 

Best regards, 

Suzanne 

Suzanne De Bruijn, PhD 

Associate Editor

PLOS Medicine

sbruijn@plos.org

Comments from the academic editor:

Please provide a more in-depth discussion on the translation of these results to other contexts. 

Comments from the reviewers: 

Reviewer #1: Thanks for the opportunity to read your manuscript. My role is statistical reviewer, so I have focused on the design, data, and analysis that are presented. I have put general comments first, followed by questions relevant to a specific section of the manuscript (with a page/line reference). 

This manuscript presents a parallel RCT testing if a 'pay it forward' strategy would increase HPV vaccine uptake in girls 15-18 in Western China. A random selection of participants registered at community health centres was taken, and then they were invited to take part in the study. Participants were included if they were able to have the HPV vaccine but had not history of receiving it. They were randomly allocated to usual care (invitation to receive a HPV vaccine and provided with information about the vaccines and prices) or the 'pay it forward' arm (usual care, plus a postcard message, and the cost of half of the first dose subsidised by previous participants). Clinical staff and assessors were blinded to the allocation. The primary study outcome was uptake of the first dose within 3 months of receipt of the intervention, and several other secondary outcomes were included (e.g. second dose uptake, number donating in pay-it-forward arm). The main analyses use a linear probability model with robust error to estimate risk difference. There were clear improvements in uptake of HPV vaccine with the 'pay it forward' strategy. 

The analyses described those in the SAP part of the protocol - thank you for providing. 

P9, L189. Just wondering if the wording about timing 'within three months after receiving the intervention' could be adjusted, as one study arm is usual care, so perhaps should be 'after randomisation' or 'first contact'? 

P11, L214. When was Fisher's exact test used over Chi square?

P11, L216. A limitation of using the linear probability model (i.e. glm with Gaussian dist and robust error) to estimate risk difference is that the correction that 'sandwich' part of the error provides is sensitive to sample size, with a large sample size needed to avoid a biased SE. Can you confirm that for the main outcome, the CI for crude difference in proportion is comparable to a Newcombe CI for risk difference? (https://doi.org/10.1002/(SICI)1097-0258(19980430)17:8%3C873::AID-SIM779%3E3.0.CO;2-I Method 10/11). Alternatively, there are methods that can estimate risk differences in a multivariable model (e.g. margins command in Stata, and the margins package in R).

P15. The p-value provided for the sub-group comparisons is inconsistent with the risk difference estimates for each level of the subgroup, e.g. the CIs for sex of caregiver show a high degree of overlap, but the p-value reported is <0.001. It's not currently reported in the methods how the sub-group analyses were completed - these should be analysed by an interaction between treatment variable, and sub-group variable, with the p-value for this parameter the test of heterogeneity in treatment according to the sub-group levels. 

Table 1. The footnote has 'Data are mean (SD)' but it looks like SD is presented with a '±' symbol. I'd suggest presenting it in brackets instead. 

Reviewer #2: This manuscript reports on a well conducted trial evaluating an intervention to increase catch-up HPV vaccine uptake among Chinese girls. While the data and results merit publication, the report shows several weaknesses and imprecisions which in my opinion need to be improved.

Abstract:

Mention that eligible were unvaccinated girls, and what the setting of the study was community health centers) 

The costs in the pay-forward strategy are 28% higher: please correct the « at comparable costs », eg, «at one quarter higher costs».

Introduction:

Please mention how many doses are recommended in China at different age groups.

Line 81: «fewer doses» makes sense only if 3 doses were recommended for catch-up

Line 82: What is delayed vaccination a barrier for? please revise the sentence.

Line 86: Please revise the sentence « the belief that their girls are sexually inactive ». At age 13-14, when parents delay vaccination despite the recommendation, most girls will indeed be sexually inactive. Reproduce the cited results as closely as possible.

Line 87: « a community-engaged solution …. ». Please revise the introduction to clarify the justification to work on the catch-up. So far, the state of the art addressed reasons for delay beyond age 14, but not factors related to missing catch-up. Are there any specific data for the reasons, if not please just say that you suppose that the reasons are the same for the initial non-adherence. Then please introduce the idea of community-engagement: what is this and why do you think it may be a solution, given evidence in Chinese population?

Line 89: please provide a specific reference to the nudge terminology

Please explain what the donations were being used for and in which way. Eg, are they given to specific users (according to which rules?), or integrated in the health center budget? In absence of this information - and given the eventually small donations made (relative to the vaccine price and the subsidy claim) -, it is not clear whether this can really be called a pay-it-forward strategy.

Another point is that social inequalities are not at all addressed, although financial barriers are presented as essential and part of the lever of the evaluated strategy. Please present a few data from China.

Methods: 

Which proportion of girls living in the area or a health center usually enrols in the lists used for recruitment , are there any specific characteristics of the population doing so (lower income)?

Line 124: « An individual-based randomization approach … was appropriate. » This statement is not clear. Please clarify what would have been the alternative and justify your judgment.

Reference 24 is not complete.

Line 135: How was the purpose of the visit at the center presented to the families? Any notion of HPV, of a randomised study? Where and when did caregivers provide consent, before or during the visit?

Line 159: Please clarify: when did families in the intervention arm receive the post-card message? DUring the consultation at the center, or later via mail? Who gave or sent the postcard?

Line 175: when did girls receive the postcards? During the vaccination consultation, or before? Are these the same as those seen by caregivers during the first consultation?

Line 181: please explain what « information sheet » is to track vaccine uptake and adverse events. Wasn't this given to vaccinated girls, only?

Line 182: when did parents answer the questionnaire about their vaccine attitudes? How was the information on donation behaviour collected? From which « expense records » where the information collected?

Please describe more precisely how vaccine delay was assessed.

Results

Line 190: Why do you specify that the vaccine supply was stable ONLY during the period? If supply is an issue, then this should be mentioned in the discussion about external validity of the results, not here.

Line 192: How was the second dose organised? Please specify the circumstances for this secondary outcome….

Line 210: Please link the sample size of 160 per arm to the 80 per center that was mentioned previously.

How were vaccine confidence data analysed? Please conduct effectiveness analyses stratified by vaccine confidence levels.

Line 252: What does « successfully invite » refer to, if the response rate is 52% ? Please revise.

Also, which percentage of the no-shows were vaccinated? If all, then the 48% los should be presented as a major barrier of the intervention feasibility. Please dicuss whether the intervention can be integrated in other center visits (routine or for specific health issues).

Line 261: Please include information on baseline vaccine confidence in the participant characteristics table. If this was not assessed at baseline, please explain why and discuss the related limitation (any risk for difference observed at follow-up already existing at baseline). Also mention in Figure 3 that these are follow-up data, without adjustment for baseline characteristics. 

Line 323: please inform on the costs that increased for the healthcare provider.

Line 334: rather « encourage to get vaccinated »

Discussion 

Line 340: please make clear that the results pertain to catch-up vaccine uptake and provide a short characterisation of the study setting

Line 342: As no information is available on baseline vaccine confidence, it is not possible to claim that the intervention was associated to higher confidence. Also, the costs are not comparable. Please revise.

Please provide arguments in the introduction for the qualification of the study setting as « middle-income country ». 

Please discuss in how far the trial participants represent the population of the town and selected health centers. 

Given the fact that actual donations were of small amount compared to the vaccine prices and of what was announced as community-subsidy, is this really a « pay-it-forward » strategy? Isn't is ethically worrisome to give wrong information about the subsidy?

As the authors point out more analyses (qualitative?) are needed to understand which element of the intervention had the positive impact. As it is, one cannot conclude that it is specifically the donation - it may just be the messages. Overall, this should be better captured by all parts of the manuscript, including the title (rather speak about stimulation of pro-social behaviour?). 

Please discuss whether this call for pro-social behaviour could have stimulated reactance, ie, it could have annoyed some caregivers who felt urged into vaccination. Did you conduct any evaluation of user satisfaction?

Line 350: Confusing: how can the follow-up vaccine uptake in the intervention arm be compared to national vaccine coverage data? Please revise this. Please avoid the term « vaccination rate », as it is ambiguous between coverage and uptake incidence among unvaccinated.

Line 354: Calling a study « old » is a bit clumsy? Please make it clear whether the UK study addressed a catch-up programme, as well. 

Discuss the contradiction between stratified results by household income and center D

Line 371: Please revise the statement about the replacement of public investment by pay-if-forward. If this is this the authors' wish, then it should be formulated as such.

Reviewer #3: Helpful and insightful research. A few queries- in Background - lines 84-87- do parents know if wait on HPV vaccine that they will have to pay ie gov't not offer HPV vaccine free to those 15yr and over ? Do they know how much it costs? The overview of the pay it forward model paper and appendix has one other aspect not noted in text - not in lines 89 thru 95 nor in Figure 1 in supplemental information . Appendix page 10 and 11 notes recruitment phone call told " Your girl is eligible for HPV vaccination through our project and she is lucky to be selected out of >1000 girls living in the neighborhood". ie have won a lottery chance -While equal for both arms RCT but merits some comment in paper to note that this may also have been a nudge to get HPV vaccine - "chance to win a prize" makes someone feel special and more positive- another kind of nudge- here prize is 50% opportunity for subsidized vaccine not true standard of care where no chance to win subsidy . This point could be raised in the discussion . Recruitment occurred over 8 months - was there any difference in volunteer/spontaneous non study uptake of HPV vaccine in that time period? Outcomes section - as noted above - " standard of care arm" was not really standard of care ie received a phone call and told in lottery to win subsidized vaccine- need another term. Costs - lines 227 -228- in supplemental materials page 11 & 12 not clear if telephone costs included personnel costs to make the phone call as well as telephone service charges . Results - not clear if most caregivers knew cost of HPV vaccine before trial- did they? , was that asked ? Discussion- again in line 341 refer to standard of care when these also received a phone call and a chance at a lottery to win discounted cost of vaccine - not standard of care - and furthermore the rate of vaccine uptake was 17.5% ( line 286) and this is over 2x rate reported for this age group in " real" standard of care - 8.6 %- line 102. This is a great increase albeit not sure how much cost for those phone calls beyond services costs lists and clinic time to learn not won lottery and given pamphlet. Looks like big bang for small money. Other reminder call studies have shown good impact . This is modified reminder. So yes the pay it forward did even more than this other control arm but that arm. These gains in the control arm deserve some attention too. This is where the " standard of care " issue could be raised as noted above - really control arm and could describe how differs for standard of care. This bonus uptake in control arm deserves attention in the discussion.

---

* Please upload any figures associated with your paper as individual TIF or EPS files with 300dpi resolution at resubmission; please read our figure guidelines for more information on our requirements: http://journals.plos.org/plosmedicine/s/figures. While revising your submission, please upload your figure files to the PACE digital diagnostic tool, https://pacev2.apexcovantage.com/. PACE helps ensure that figures meet PLOS requirements. To use PACE, you must first register as a user. Then, login and navigate to the UPLOAD tab, where you will find detailed instructions on how to use the tool. If you encounter any issues or have any questions when using PACE, please email us at PLOSMedicine@plos.org.

* Please ensure that the study is reported according to the CONSORT guideline and include the completed CONSORT checklist as Supporting Information. When completing the checklist, please use section and paragraph numbers, rather than page numbers. Please add the following statement, or similar, to the Methods: "This study is reported as per CONSORT guideline (S1 Checklist)."

* Abstract: Please structure your abstract using the PLOS Medicine headings (Background, Methods and Findings, Conclusions). In the last sentence of the Abstract Methods and Findings section, please describe the main limitation(s) of the study's methodology.

FIGURES AND TABLES

SUPPLEMENTARY MATERIAL

REFERENCES

Randomized Controlled Trials 

* PLOS Medicine requires that all trials be prospectively registered in one of registries recognized by WHO. Please ensure that study registration details are included in the Methods section.

* Please structure the Methods section using the following sub-headings: Study design and participants, Randomization and masking, Procedures, Outcomes, Statistical analysis.

* Please ensure that all prespecified outcomes (primary, secondary, and exploratory) are listed in the Methods/Outcomes section and indicate whether there are outcomes that are not presented in the current report.

* Please specify the dates (Month Day, Year) during which study enrollment and follow up occurred.

* Please include absolute numbers wherever you report percentages; eg, n/N (%)

* Please present the safety data for the study including numbers of specific events and whether or not adverse events are thought to be related to treatment. AEs should be reported in the abstract, per CONSORT and CONSORT-Harms.

* Please complete the CONSORT checklist (https://www.equator-network.org/reporting-guidelines/consort/) and ensure that all components of CONSORT are present in the manuscript, including how randomization was performed, allocation concealment, blinding of intervention, definition of lost to follow-up, power statement. When completing the checklist, please use section and paragraph numbers, rather than page numbers.

* Please report your abstract according to CONSORT for abstracts, following the PLOS Medicine abstract structure (Background, Methods and Findings, Conclusions) https://www.equator-network.org/reporting-guidelines/consort-abstracts/

* If your trial had to undergo important modifications in response to extenuating circumstances, please complete the CONSERVE-CONSORT checklist and provide in your Supporting Information; (https://www.equator-network.org/reporting-guidelines/guidelines-for-reporting-trial-protocols-and-completed-trials-modified-due-to-the-covid-19-pandemic-and-other-extenuating-circumstances-the-conserve-2021-statement/). When completing the checklist, please use section and paragraph numbers, rather than page numbers.

* In keeping with our commitment to Open Science, please include the study protocol document and analysis plan (including any amendments) as Supporting Information to be published with the manuscript if accepted.

* Please note that PLOS Medicine requires prospective, public registration of a data sharing plan (as part of mandatory clinical trials registration) for all clinical trials that began enrollment on or after January 1, 2019, in accordance with ICMJE requirements.

---

## [Decision Letter · Decision Letter 2]

11 Jun 2025

Dear Dr. Wu,

Thank you very much for re-submitting your manuscript "Pay-it-forward strategy reduced HPV vaccine delay and increased uptake among catch-up age girls: A randomized clinical trial" (PMEDICINE-D-25-00007R2) for review by PLOS Medicine.

I have discussed the paper with my colleagues and the academic editor and it was also seen again by 2 reviewers. I am pleased to say that provided the remaining editorial and production issues are dealt with we are planning to accept the paper for publication in the journal.

[LINK]

We also have several editorial requests, which you can also find at the end of this email.

We look forward to receiving the revised manuscript by Jun 18 2025 11:59PM.   

Sincerely,

Suzanne De Bruijn, PhD

Associate Editor 

PLOS Medicine

plosmedicine.org

Requests from Editors:

*Please change your title, to be non-declarative.

ABSTRACT:

*please specify what '54% clinic attendance rate refers to.

*Please provide details about randomisation

*Please provide details abut study setting

*Please specify who was blinded to the intervention and control.

*Please state that analysis was intention to treat.

*Please provide the number of participants lost to follow up in each group.

*Please indicates the dates during which study enrollment and follow up occurred.

*Please clarify that the prices that are mentioned for the vaccine in both arms were not actually what was paid. Please ensure this is clear throughout the manuscript. 

*Please include 'Chinese' in the last sentence of the Methods and Findings paragraph (in “conduct in a single western province”).

Statements:

*please remove funding statement from the main text (in the Method sections)

*Please remove the funding statement from the main text, at the end of the discussion (statement starting on line 442).

*please include a statement on code in DAS.

Checklist:

*as your study was affected by COVID, please fill out the CONSERVE-CONSORT checklist. 

*Please ensure that the checklists mentions paragraphs and sections rather than line numbers.

* Please confirm that your title complies with to PLOS Medicine's style. Your title must be nondeclarative and not a question. It should begin with main concept if possible. "Effect of" should be used only if causality can be inferred, i.e., for an RCT. Please place the study design ("A randomized controlled trial," "A retrospective study," "A modelling study," etc.) in the subtitle (ie, after a colon).

* Please confirm that your abstract complies with our requirements, including format (three sections: Background, Methods and Findings, and Conclusions) and providing all the information relevant to this study type https://journals.plos.org/plosmedicine/s/submission-guidelines#loc-abstract

* Please ensure that all abbreviations are defined at first use throughout the text.

* Please confirm that all numbers presented in the abstract are present and identical to numbers presented in the main manuscript text.

* In the author summary, in the final bullet point of 'What Do These Findings Mean?', please include the main limitations of the study in non-technical language.

* The funding statement should include: specific grant numbers, initials of authors who received each award, URLs to sponsors’ websites. Also, please state whether any sponsors or funders (other than the named authors) played any role in study design, data collection and analysis, the decision to publish, or preparation of the manuscript. If they had no role in the research, include this sentence: “The funders had no role in study design, data collection and analysis, decision to publish, or preparation of the manuscript.

* All authors must declare their relevant competing interests per the PLOS policy, which can be seen here: https://journals.plos.org/plosmedicine/s/competing-interests For authors with ties to industry, please indicate whether any of the interests has a financial stake in the results of the current study.

* Please add this statement to the manuscript's Competing Interests: "EHG is an Academic Editor on PLOS Medicine's editorial board.

*We appreciate that you provide all the data in an Excel file. However, it appears that you have included data that may breach patient confidentiality. Please edit your data set in accordance with patient consent, and remove the following personal data: birthdates

*Please also labels all columns in this excel file properly, so the file is understandable for readers.

Comments from Reviewers:

Reviewer #1: Thanks for the revised manuscript and responses to my original review. Most of the updates have resolved my initial review.

Figure 2 is a good representation of the sub-group analysis, but the reporting of this in the methods needs revision. The interaction p-value is what provides evidence for heterogeneity - in this case it was only awareness of HPV vaccinations which had heterogeneity. Making inferences about when sub-groups meets significance (or had a confidence limit that crosses the null threshold) is called 'differences in nominal significance' and is creates a liberal bias in inference, e.g. in many of these subgroups the CL cross. For this section, the main result is just for awareness of HPV vaccinations, with none of the rest of the subgroups showing evidence of heterogeneity. 

Reviewer #2: Thank you for the rebuttal, which answers all of my questions.

My preference would be that your clear answers would lead to some clarification in the manuscript on ALL points - assuming that if one aspect was not clear to me, it may not be clear to readers, either. This concerns questions 9, 10, 13, 22, 24, 34 and 37. 

For Question 31, I rather was thinking of external validity of the trial results, not of selection bias. Your trial results are certainly internally valid, indepent of the selection of participants. However, they may not be applicable to any resident, if participants were highly selected.

[LINK]

---

## [Editor Report · Decision Letter 3]

9 Jul 2025

Dear Dr. Wu,

Thank you very much for re-submitting your manuscript "Effect of pay‑it‑forward strategy to reduce HPV vaccine delay and increase uptake among catch-up age girls: A randomized controlled trial" (PMEDICINE-D-25-00007R3).

We appreciate the revisions you made, but we have several remaining requests:

1) We suggest to change your title to “Effect of a pay-it-forward strategy on reducing HPV vaccine delay and increasing uptake among 15- to 18-year-old girls in China: A randomized controlled trial”

2) Please include URLs in the funding statement

3) Please mention the CONSERVE-CONSORT checklist in the main text (with a reference to the checklist), and state why you completed it (e.g. study affected by the pandemic).

4) Please define HPV the first time you mention it in the abstract and the main text.

5) The excel file still contains IDs; please remove these. Please also consider providing a second tab with more information; e.g. what 0 and 1 for arm means.

6) Thank you for including a code statement in the Data Availability Statement (DAS). Please ensure that the DAS in your online submission is identical to the one in the manuscript itself.

7) Please clearly state that the first vaccine was for free for the intervention arm (rather than stating subsidized, or giving a RMB number.)

8) Please use only $ or RMB in the abstract, not both. We are happy if you want to mention both in the methods section.

9) Abstract: thank you for including the data we ask for. However, it now reads slightly confusing. We suggest you rewrite so that the flow goes:

1) We identified 662 participants per phone invitation

2) only 360 showed up in the health centra (this is the number lost to follow up)

3) they then were assigned to experimental or control arm, and unblinded.

4) treatment was ‘pay-it-forward’ or control, primary outcome was first HPV vaccine.

10) Abstract: Consider adding a second sentence to the Conclusions, for instance that the participants in the pay-it-forward method also showed higher vaccine confidence. 

We look forward to receiving the revised manuscript by Jul 16 2025 11:59PM.   

Sincerely,

Suzanne De Bruijn, PhD

Associate Editor 

PLOS Medicine

plosmedicine.org

---

## [Editor Report · Decision Letter 4]

16 Jul 2025

Dear Dr Wu, 

On behalf of my colleagues and the Academic Editor, Rebecca Grais, I am pleased to inform you that we have agreed to publish your manuscript "Effect of a pay‑it‑forward strategy on reducing HPV vaccine delay and increasing uptake among 15- to 18-year-old girls in China: A randomized controlled trial" (PMEDICINE-D-25-00007R4) in PLOS Medicine.

We have 2 minor remaining requests, we like you to address:

-Please change “for this trial affected by the COVID-19 pandemic” to “for this trial was affected by the COVID-19 pandemic”

-Please include in the abstract that the subsidy of $47.7 covers the first vaccine.

PRESS

Sincerely, 

Suzanne De Bruijn, PhD 

Associate Editor 

PLOS Medicine